# Overexpression of a Cytosolic 6-Phosphogluconate Dehydrogenase Gene Enhances the Resistance of Rice to *Nilaparvata lugens*

**DOI:** 10.3390/plants9111529

**Published:** 2020-11-10

**Authors:** Lin Chen, Peng Kuai, Miaofen Ye, Shuxing Zhou, Jing Lu, Yonggen Lou

**Affiliations:** State Key Laboratory of Rice Biology & Ministry of Agriculture Key Lab of Molecular Biology of Crop Pathogens and Insects, Institute of Insect Sciences, Zhejiang University, Hangzhou 310058, China; chenlin88@126.com (L.C.); kpchen7493@163.com (P.K.); 11616070@zju.edu.cn (M.Y.); 3090100232@zju.edu.cn (S.Z.); jing_lu@zju.edu.cn (J.L.)

**Keywords:** rice, *6-phosphogluconate dehydrogenase*, herbivore-induced plant defenses, *Nilaparvata lugens*

## Abstract

The pentose phosphate pathway (PPP) plays an important role in plant growth and development, and plant responses to biotic and abiotic stresses. Yet, whether the PPP regulates plant defenses against herbivorous insects remains unclear. In this study, we cloned a rice cytosolic 6-phosphogluconate dehydrogenase gene, *Os6PGDH1*, which encodes the key enzyme catalyzing the third step in the reaction involving the oxidative phase of the PPP, and explored its role in rice defenses induced by brown planthopper (BPH) *Nilaparvata lugens*. Levels of *Os6PGDH1* transcripts were detected in all five examined tissues, with the highest in outer leaf sheaths and lowest in inner leaf sheaths. *Os6PGDH1* expression was strongly induced by mechanical wounding, infestation of gravid BPH females, and jasmonic acid (JA) treatment. Overexpressing *Os6PGDH1* (oe6PGDH) decreased the height of rice plants and the mass of the aboveground part of plants, but slightly increased the length of plant roots. In addition, the overexpression of *Os6PGDH1* enhanced levels of BPH-induced JA, jasmonoyl-isoleucine (JA-Ile), and H_2_O_2_, but decreased BPH-induced levels of ethylene. Bioassays revealed that gravid BPH females preferred to feed and lay eggs on wild-type (WT) plants over oe6PGDH plants; moreover, the hatching rate of BPH eggs raised on oe6PGDH plants and the fecundity of BPH females fed on these were significantly lower than the eggs and the females raised and fed on WT plants. Taken together, these results indicate that *Os6PGDH1* plays a pivotal role not only in rice growth but also in the resistance of rice to BPH by modulating JA, ethylene, and H_2_O_2_ pathways.

## 1. Introduction

Upon attack by herbivores, plants recognize herbivore-associated molecular patterns via cell-membrane-localized receptors and then initiate early signaling events, such as the activation of mitogen-activated protein kinase (MAPK) cascades and the burst of reactive oxygen species (ROS) [1]. These early signaling events activate signaling pathways mediated by phytohormones consisting mainly of jasmonic acid (JA), salicylic acid (SA), and ethylene [2], which in turn alter the transcriptome and metabolome of plants and increase the resistance of plants to herbivores [3,4].

The pentose phosphate pathway (PPP), which exists widely in plants and is one of the essential cellular metabolic pathways, is closely linked to glycolysis via shared common substrates, such as glucose-6-phosphate, fructose-6-phosphate, and glyceraldehyde-3-phosphate [5,6]. By providing reductants in the form of nicotinamide adenine dinucleotide phosphate (NADPH) for anabolic metabolism, maintaining carbon homoeostasis and supplying carbon skeletons for synthesis of nucleic acids, amino acids, and phenylpropanoids, the PPP plays an important role in plant growth and development [5,7]. Moreover, by maintaining the redox potential necessary to protect against oxidative stress, the PPP is involved in the defense responses of plants against biotic and abiotic stresses [8], including salt and UV-B radiation [9], drought [10], cold [11], aluminum [12], and pathogen infection [13]. In soybean, for example, a cytosolic glucose-6-phosphate dehydrogenase (G6PDH, EC1.1.1.49), one of the key enzymes in the PPP which oxidizes glucose-6-phosphate to produce 6-phosphogluconolactone and NADPH, GmG6PDH2 promotes the resistance of soybean to salt stress by suppressing the salinity-induced generation of reactive oxygen species [14]. In *Nicotiana tabacum*, the overexpression of an engineered G6PDH in the cytosol of a *Phytophthora nicotianae*-susceptible tobacco cultivar enhances pathogen resistance and abiotic stress tolerance by regulating early oxidative bursts, callose deposition, and defense-related metabolic source-to-sink transitions [15]. In addition, Hu et al. [13] reported that *g6pd5/6*, a double mutant with loss of function of the two cytosolic isoforms of G6PDHs, G6PD5, and G6PD6 in Arabidopsis, showed increased susceptibility to root knot nematodes due to the suppression of ROS production and the decrease in transcript levels of WRKY transcription factors and SA-responsive and JA-dependent genes. In summary, while the PPP seems to play a central role in many plant processes, including plant disease resistance, its role in herbivore-induced plant defenses remains almost entirely unclear.

The 6-phosphogluconate dehydrogenase (6PGDH, EC 1.1.1.44), which catalyzes the oxidative decarboxylation of 6-phosphogluconat and produces ribulose-5-phosphate concomitant with the generation of NADPH [5,8], is one of the key enzymes in the PPP [5]. In plants, 6PGDH presents in both cytosol and plastids [16]. To date, genes encoding 6PGDHs have been cloned and characterized in some plant species, including Arabidopsis [17], maize [18], tomato [19], and spinach [20]. In Arabidopsis, there are three known 6PGDH genes, one encoding a cytosolic 6PGDH and two encoding plastidic 6PGDHs [20]; in rice, there are at least one plastidic 6PGDH gene and one cytosolic 6PGDH gene [21]. It has been reported that 6PGDH genes play an important role in plant growth, development, and responses to multiple stresses [17,22]. For instance, in Arabidopsis, *AtPGD2* is required for the female gametophyte to provide guidance to ovules when establishing the pollen tube, and mutants lacking *AtPGD2* are homozygous lethal [17]. Potato virus Y markedly increases the activity of 6PGDH in tobacco leaf tissues [23], and an elicitor from yeast cell walls induces the expression of 6PGDH in alfalfa suspension cells [24]. Moreover, in rice, abiotic stresses, such as drought, cold, and high salinity, abscisic acid treatment, as well as infection by pathogens, such as sheath blight fungus *Rhizoctonia solani*, enhance transcript levels or the activity of 6PGDHs [21,25]. However, whether and how 6PGDH genes regulate the defenses of plants against herbivores remain largely unknown.

Therefore, we cloned a rice cytosolic 6-phosphogluconate dehydrogenase, designated as *Os6PGDH1*, and elucidated its role in herbivore-induced defenses in rice, the most important food crop in the world. Previous studies have revealed that in response to infestation by herbivores, including the brown planthopper (BPH) *Nilaparvata lugens*, rice plants activate MAPK cascades and WRKY transcription factors, as well as change the biosynthesis of defense-related signal molecules, such as JA, jasmonoyl-isoleucine (JA-Ile), SA, ethylene, and H_2_O_2_; these changes subsequently cause the production of defense responses of rice against herbivores [26,27,28,29]. In this study, we found that the overexpression of *Os6PGDH1* increases the BPH-induced accumulation of JA, JA-Ile, and H_2_O_2_, but decreases ethylene levels, resulting in the enhanced resistance of rice to BPH. This suggests that the *Os6PGDH1-*mediated PPP plays a role in the resistance of rice to herbivores.

## 2. Results

### 2.1. cDNA Cloning and Phylogenetic Analysis of the Os6PGDH1 Sequence

The full-length cDNA of *Os6PGDH1* was cloned via reverse transcription PCR from a cDNA library of rice variety Xiushui 110 (wild-type, WT; Appendix A), which is 1443 base pairs in length and encodes a 6-phosphogluconate dehydrogenase protein of 480 amino acids (Appendix A). The Os6PGDH1 protein was predicted to have a nicotinamide adenine dinucleotide phosphate (NADP)-binding site (NAD_binding_2) in the N-terminus and a 6-phosphogluconate (6PD)-binding site (6PGD) in the C-terminus (Appendix A). To assess the relationship between *Os6PGDH1* and its homologous proteins from other plant species, a phylogenetic tree was constructed based on amino acid sequences of 18 6PGDHs. The phylogenetic tree showed that rice Os6PGDH1 is closely related to Si6PGDH1 in *Setaria italica*, Sb6PGDH1 in *Sorghum bicolor*, Zm6PGD1 and Zm6PGD2 in *Zea mays*, and Bd6PGDH1 in *Brachypodium distachyon*; they share 96.04%, 95.43%, 93.98%, 95.00%, and 93.54% identity, respectively (Figure 1).

### 2.2. Expression of Os6PGDH1

Transcriptional analysis revealed that *Os6PGDH1* was widely expressed in all rice tissues, with the highest expression level in outer leaf sheaths, followed by roots and leaves. In comparison to other tissues, inner leaf sheaths and seeds showed the lowest expression of *Os6PGDH* transcripts (Figure 2).

We also investigated the transcript level of *Os6PGDH1* in rice plants treated with mechanical wounding, JA, or SA, or infested with BPH. The result showed that mechanical wounding (1.5 to 24 h post-treatment), BPH infestation (3 to 48 h post-treatment), and JA treatment (1.5, 6 to 48 h post-treatment) all induced the expression of *Os6PGDH1* (Figure 3A–C). In addition, SA treatment slightly induced the expression of *Os6PGDH1* (Figure 3D). These data suggest that the transcript level of *Os6PGDH1* varied with rice tissues and that *Os6PGDH1* may be involved in herbivore-induced defenses in rice.

### 2.3. Overexpressing of Os6PGDH1

To explore the role of *Os6PGDH1* in BPH-induced rice defenses, we obtained two T_2_ homozygous lines overexpressing *Os6PGDH1* (oe6PGDH lines), oe-15 and oe-39, each with a single insertion (Figure 4A). Transcript analysis revealed a significant increase in basal and induced transcript levels of *Os6PGDH1* in oe6PGDH lines at 12 h post mechanical wounding or infestation with gravid BPH females compared to those in equally treated WT plants (Figure 4B–D), suggesting that *Os6PGDH1* gene was successfully overexpressed in oe6PGDH lines.

Compared to WT plants, oe6PGDH plants exhibited lower height and mass of aboveground parts (Figure 5A–C). However, the root length increased in oe6PGDH plants compared to WT plants, although no significant difference in root length was observed between oe-15 plants and WT plants (Figure 5D). The results indicate that *Os6PGDH1* regulates rice growth and development.

### 2.4. Overexpression of Os6PGDH1 Enhances Levels of BPH-Induced JA, JA-Ile, and H_2_O_2_, but Suppresses Ethylene

It has been well documented that JA, JA-Ile, SA, ethylene, and H_2_O_2_ play a key regulatory role in defense responses of rice against BPH [26,27,30]. To test whether *Os6PGDH1* influences the biosynthesis of signal molecules, we examined the levels of these molecules in transgenic and WT plants post-infestation with gravid BPH females. Similar to previous findings [31,32], infestation with gravid BPH females induced the production of JA and JA-Ile during the whole tested period (Figure 6A,B). Overexpression of *Os6PGDH1* enhanced the BPH infestation-induced accumulation of JA (3, 8, and 48 h post-infestation) and JA-Ile (3, 8, and 48 h), although such phenomena were observed in one oe6PGDH line only at certain time points (Figure 6A,B). Overexpressing *Os6PGDH1* also increased BPH-elicited levels of H_2_O_2_ at 12, 24, and 48 h after infestation (Figure 6C). In contrast, levels of ethylene emitted from oe6PGDH plants were significantly lower than those from WT plants at 24, 48, and 72 h post-infestation (Figure 6D). BPH-induced SA levels in oe6PGDH lines were similar to those in WT plants (Appendix A). These results suggest that in rice, *Os6PGDH1* is involved in the biosynthesis of BPH-induced JA, JA-Ile, H_2_O_2_, and ethylene, but not SA.

### 2.5. Os6PGDH1 Positively Regulates the Resistance of Rice to BPH

Since overexpressing *Os6PGDH1* alters defense-related signaling pathways, we asked if the overexpression of *Os6PGDH1* affects the resistance of rice to BPH. Bioassays revealed that gravid BPH females preferred to feed and oviposit on WT plants rather than on oe6PGDH plants, suggesting that oe6PGDH plants had an antixenosis effect on BPH females (Figure 7A,B). Moreover, the hatching rate of BPH eggs was lower in transgenic lines than in WT plants (Figure 7C). Although oe6PGDH lines had no effect on survival rate of BPH nymphs (Appendix A), the number of eggs laid by a BPH female adult emerged from oe6PGDH lines was significantly lower than that laid by a female adult emerged from WT plants (Figure 7D). These results suggest that overexpressing *Os6PGDH1* enhanced the resistance of rice to BPH.

## 3. Discussion

In this study, we cloned the rice cytosolic 6-phosphogluconate dehydrogenase *Os6PGDH1* and investigated its role in plant growth and BPH-induced defense responses. We found that *Os6PGDH1* was mainly expressed in outer leaf sheaths, leaves, and roots of rice plants, and was strongly induced by mechanical wounding, gravid BPH female infestation, and JA treatment (Figure 2 and Figure 3). The overexpression of *Os6PGDH1* impaired the growth of aboveground parts of plants but slightly promoted the growth of roots (Figure 5). Moreover, overexpressing *Os6PGDH1* enhanced BPH-induced levels of JA, JA-Ile, and H_2_O_2,_ and decreased the level of ethylene (Figure 6), which in turn conferred the resistance of rice to BPH (Figure 7). Our findings demonstrate that *Os6PGDH1*, in addition to its role in plant development and responses to abiotic stresses and pathogen infection, plays an important role in regulating the resistance of rice to herbivores.

In higher plants, 6PGDH can be divided into cytosolic and plastidic isoforms [16]. We found that Os6PGDH1 has a more closely evolutionary relationship with 6PGDHs of *Setaria italica*, *Sorghum bicolor*, and *Zea mays*, all of which were cytosolic isoforms [18,33,34], confirming that Os6PGDH1 is a cytosolic 6PGDH (Figure 1). *Os6PGDH1* was widely expressed in rice tissues but mainly expressed in leaves, outer leaf sheaths, and roots, tissues that are active in anabolic metabolism (Figure 2); moreover, *Os6PGDH1* was induced by mechanical wounding, infestation with gravid BPH females, and JA treatment (Figure 3). This expression pattern of the *6PGDH* gene, as one of the genes encoding key enzymes in the PPP that provides reductants and precursors for anabolic metabolism and maintains the redox potential necessary to protect against oxidative stress, fits its function well. Interestingly, we found that the expression of *Os6PGDH1* was induced by SA treatment at 1.5 h (Figure 3D). Future research should perhaps elucidate this reason.

Factors that caused growth retardation of oe6PGDH lines might be the decreased NADP^+^ level or the increased NADPH/NADP^+^ ratio, the decrease of substrate availability from glycolysis for plant growth, the trade-off between plant growth and defense, and/or the combination of factors as mentioned above. It has been reported that low NADP^+^ levels or high NADPH/NADP^+^ ratios impair plant growth [35]. 6PGDH catalyzes the third step in the PPP’s oxidative phase using NADP^+^ as a cofactor to generate NADPH [5]. Thus, it is conceivable that the overexpression of *Os6PGDH1* decreases the NADP^+^ level and increases the NADPH/NADP^+^ ratio. The PPP is linked to glycolysis [5,6]; hence, overexpressing *Os6PGDH1* may weaken glycolysis, resulting in incapacity to metabolize carbohydrates properly for growth. It has been well documented that there is a trade-off between plant growth and defense [36,37]. Given that the overexpression of *Os6PGDH1* significantly enhances the resistance of rice to BPH, it is reasonable to think that some compounds for plant growth were used to synthesize defensive compounds in oe6PGDH plants. Further research should elucidate the mechanism underlying Os6GPDH1-mediated plant growth. 

Many studies have demonstrated that the PPP plays a crucial role in regulating levels of JA and H_2_O_2_ in plants by influencing the level of NADPH, a reductant required for the biosynthesis of these signal molecules [38]. For instance, in tobacco cultivar Xanthi, cryptogein, an elicitor secreted by the oomycete *Phytophthora cryptogea*, activates the PPP, which produces NADPH for plasma membrane oxidases to generate reactive oxygen species (ROS) [39]. In Arabidopsis, the double mutant *g6pd5/6* that loses cytosolic G6PDH activity, reduced root-knot nematode infection-induced ROS production, as well as the expression of WRKY genes, JA-dependent gene *PDF1.2*, and ROS-generating NADPH oxidase coding genes *RbohD* and *RbohF* [13]. Wang et al. [40] found that the inhibition of G6PDH activity by G6PDH inhibitor glucosamine decreased salt-induced NADPH content and NADPH oxidase activity, which repressed the accumulation of salt-induced H_2_O_2_ in *Artemisia annua*; likewise, soybean cytosolic G6PDH activity is responsible for aluminum-triggered total ROS accumulation [12]. In Arabidopsis, 12-oxophytodienoate reductase 3 (OPR3) catalyzes the reduction of (-)-cis-12-oxophytodienoic acid (OPDA) to JA natural precursor (+)-cis-OPDA; during this process, NADPH is specifically required [41]. Therefore, the increase in levels of JA, JA-Ile, and H_2_O_2_ in oe6PGDH lines might be related to the increased level of NADPH produced by Os6PGDH1 (Figure 6). Interestingly, we found that overexpressing *Os6PGDH1* decreased BPH-induced levels of ethylene in rice (Figure 6D). Whether the decrease in levels of ethylene is also related to levels of NADPH or glycolysis remains to be investigated in the future.

JA, H_2_O_2_, and SA signaling pathways have been reported to positively regulate the resistance of rice to BPH [27,42], whereas the ethylene-mediated signaling pathway negatively regulates its resistance to BPH [26,43]. Hence, the enhanced resistance in oe6PGDH plants to BPH may at least in part result from high levels of JA, JA-Ile, and H_2_O_2_ as well as low levels of ethylene. Which compounds regulated by these changed signaling pathways and/or Os6PGDH1 influence the performance of BPH should be elucidated in the future. 

In summary, our study demonstrated that Os6PGDH1 plays an important role in rice growth and resistance to BPH. When attacked by BPH, rice plants perceive the signals from the herbivore and activate the expression of *Os6PGDH1*, which may in turn produce NADPH to promote the biosynthesis of JA, JA-Ile, and H_2_O_2_; these changes influence the production of other signal molecules, such as ethylene, thereby regulating the resistance of rice to BPH. Simultaneously, the activation of Os6PGDH1 impairs glycolysis, which in turn retards plant growth. We propose Os6PGDH1 as an important node that regulates the trade-off between plant growth and defense.

## 4. Materials and Methods

### 4.1. Plant Growth

The rice (*Oryza sativa*) genotypes used in this study were Xiushui 110 (WT) and *Os6PGDH1*-overexpressing lines (oe6PGDH lines, see below). The seeds of WT and transgenic lines were soaked in water for 2 d and then cultured in a growth chamber with a 14/10 h light/dark cycle at 28 ± 2 °C. One week later, the seedlings were transplanted to 30 L hydroponic boxes (length, 50 cm; width, 35 cm; height, 17 cm) containing a rice nutrient solution [44]. After three weeks, healthy seedlings were individually transferred to 500 mL hydroponic plastic pots (diameter 8.4 cm, height 11.4 cm) and used for experiments 4–5 days later.

### 4.2. Insects

The laboratory population of BPH used in this study was originally collected from the rice fields in Hangzhou, China, and reared on BPH-susceptible rice cultivar Taichung Native 1 (TN1) in a climate-controlled room at 26 ± 2 °C, with 65% humidity and a 14 h light/10 h dark cycle.

### 4.3. Isolation and Characterization of Os6PGDH1

The full-length coding sequence of *Os6PGDH1* (accession no. LOC_Os06g02144) was PCR-amplified from a cDNA library of leaf sheaths of Xiushui 110 using the specific primers Os6PGDH1-F1 (5′-GGGGTACCATGGCTGTCACTAGAATTG-3′) and Os6PGDH1-R1 (5′-GCTCTAGATCACATCTTAGCAGCACGC-3′) that incorporated *Kpn* Ι and *Xba* I restriction sites (underlined sequences) at the 5′ and 3′ ends. The primers were designed according to the sequences on the Rice Genome Annotation Project database (http://rice.plantbiology.msu.edu/). The PCR products were ligated to the p*EASY*-Blunt Simple Cloning Vector (TransGen, Beijing, China) following the manufacturer’s instructions and sequenced to yield the pOs6PGDH1 construct.

### 4.4. Sequence and Phylogenetic Analysis of Os6PGDH1

The putative conserved domains of the Os6PGDH1 protein were analyzed using SMART (simple modular architecture research tool; http://smart.embl-heidelberg.de/). The Os6PGDH1 homologs in other plant species were identified by NCBI BLASTP (https://blast.ncbi.nlm.nih.gov/Blast.cgi) using the full-length amino acid sequence of Os6PGDH1 as a query with default parameters. Then, the amino acid sequences, which were highly homologous with the Os6PGDH1, were downloaded from the NCBI website, and multiple sequence alignment was carried out using ClustalW program in Mega X [45]. A neighbor-joining phylogenetic tree was constructed by MEGA X with 1000 bootstrap replicates using the default parameters as described in reference [45].

### 4.5. Generation and Characterization of Transgenic Plants

The pOs6PGDH1 construct and binary vector pCAMBIA1301 were digested with *Kpn* Ι and *Xba* I restriction enzymes, and then the *Os6PGDH1* fragment was inserted into *Kpn* Ι and *Xba* I restriction sites of pCAMBIA1301. The resulting overexpression transformation vector pCAMBIA1301:*Os6PGDH1* was digested by *Kpn* Ι and *Xba* I restriction enzymes to verify that the *Os6PGDH1* fragment had been correctly inserted into the vector pCAMBIA1301 (Appendix A). The expressing vector was introduced into *Agrobacterium tumefaciens* strain EHA105 and then transformed into Xiushui 110 plants by *Agrobacterium*-mediated transformation [46]. The screening of T_2_ homozygous oe6PGDH lines and the identification of the number of insertions were performed using the same method as described in Zhou et al. [26]. Two T2 homozygous lines, oe-15 and oe-39, each with a single insertion, were used for further analyses.

### 4.6. Plant Treatments

For the mechanical wounding treatment, the lower portion of rice plants (about 4 cm long) was individually pricked with a fine needle 200 times. Nonmanipulated plants were used as controls (Con). For BPH treatment, plants were individually infested with 15 gravid BPH females that were confined in a cylindrical glass cage (diameter 4 cm, height 8 cm, with 48 small holes, diameter 0.8 mm). Plants with an empty cylindrical glass cage were used as controls (noninfested). For JA or SA treatment, rice plants were individually sprayed with 2 mL of JA (100 µg mL^−1^) or SA (70 µg mL^−1^) solution in 50 mM sodium phosphate buffer; control plants were sprayed with the buffer (BUF).

### 4.7. Quantitative Real-Time PCR

Total RNA was extracted from various samples using the TaKaRa MiniBEST Plant RNA Extraction Kit (Takara, Dalian, China), according to the manufacturer’s instructions. An amount of 1 μg of total RNA from each sample was reverse-transcribed to the first-strand cDNAs using PrimeScript™ RT Master Mix Kit (TaKaRa, Dalian, China), according to the manufacturer’s protocols. qRT-PCR was performed using the Premix Ex Taq™ Kit (TaKaRa, Dalian, China) on a CFX96TM Real-Time System (Bio-Rad, Hercules, CA, USA), and the rice actin gene *OsActin* (accession no. LOC_Os03g50885) was used as an internal control. A linear standard curve, threshold cycle number versus log_10_ value (designated transcript level), was prepared using a fivefold serial dilution of a specific cDNA standard, and the relative expression levels for genes of target were calculated using the standard curve. Primers and probes used for qRT-PCR are listed in Appendix A.

### 4.8. Os6PGDH1 mRNA Expression Analysis

To examine mRNA levels of *Os6PGDH1* in different tissues of plants, tissues of WT plants, including leaves (the fifth fully expanded leaves), inner leaf sheaths, outer leaf sheaths, and roots of 30-day-old seedlings (indicated in Figure 2a), and seeds were harvested. Total RNA was extracted for all samples, after which *Os6PGDH1* mRNA expression levels were detected by qRT-PCR. 

To analyze the effects of various stresses on mRNA levels of *Os6PGDH1*, WT plants were randomly assigned to one of seven treatments: JA, SA, BUF, MW, BPH, noninfested, and Con. Transcript levels of *Os6PGDH1* in outer leaf sheaths of plants, harvested at 0, 0.5, 1.5, 3, 6, 12, 24, and 48 h post-treatment, were measured by qRT-PCR. To determine the efficiency of overexpression, *Os6PGDH1* mRNA levels in outer leaf sheaths of oe6PGDH lines and WT plants (basal and 12 h after they were mechanically wounded or infested with gravid BPH females) were detected by qRT-PCR.

### 4.9. Measurement of Plant Growth Parameters

To evaluate plant growth phenotype, plant growth parameters, including plant height, root length, and mass of aboveground part of a plant of oe6PGDH and WT lines, were measured at 28 days after plant germination. Plant height and root length were defined as the part of a plant from the stem base to the longest leaf apex and the part from the stem base to the longest root tip, respectively. Plants were cut off just at the base of the stem, and the mass of the aboveground part of plants was measured. Twenty-five individual plants of each line were used for the measurement.

### 4.10. JA, JA-Ile, and SA Analysis

WT and oe6PGDH plants were randomly assigned to BPH and noninfested treatment. Outer leaf sheaths of plants were harvested at 0, 3, 8, 12, 24, and 48 h after infestation with 15 gravid BPH females. Samples were ground in liquid nitrogen, and JA, JA-Ile, and SA were extracted by ethyl acetate containing labeled internal standards (D_6_-JA, D_6_-JA-Ile, and D_4_-SA) and analyzed using an HPLC/mass spectrometry/mass spectrometry system as described in [47]. Six biological replications were used for each treatment at each time point.

### 4.11. Hydrogen Oxide Analysis

Plants of oe6PGDH and WT lines were randomly assigned to BPH and noninfested treatment. Outer leaf sheaths of plants were harvested at 0, 3, 8, 12, 24, and 48 h after infestation with 15 gravid BPH females. Samples (about 0.1 g) were ground in liquid N_2_, and each was mixed with 1 mL of cold sterile double-distilled water. Samples were centrifuged at 13,000 rpm for 10 min at 4 °C and then the supernatants were collected. The hydrogen peroxide concentrations in the supernatants were determined using an Amplex^®^ Red Hydrogen Peroxide/Peroxidase Assay Kit (Invitrogen, Eugene, OR, USA), according to the manufacturer’s protocol. Six biological replications were used for each time point and treatment.

### 4.12. Ethylene Analysis

Plants of oe6PGDH and WT lines were randomly assigned to BPH and noninfested treatment. Each plant was covered with a sealed glass cylinder (diameter 4 cm, height 50 cm). Ethylene content was measured at 12, 24, 48, and 72 h after the start of BPH infestation using the same method as previously described in [30]. Ten biological replications were used for each treatment at each time point.

### 4.13. BPH Bioassays

To measure antixenosis effect of oe6PGDH plants on BPH females, pots containing two plants, an oe6PGDH plant and a WT plant, were individually confined in cylindrical glass cages (diameter 4 cm, height 8 cm, with 48 small holes, diameter 0.8 mm; Appendix A). Then, 15 gravid BPH females were released into each cage. The number of gravid BPH females on each plant was recorded at 1, 4, 8, 12, 24, and 48 h after the initial release, and 48 h later, the number of BPH eggs on each plant was counted under a stereoscopic microscope. The experiment was replicated 10 times.

To evaluate the impact of overexpressing *Os6PGDH1* on the hatching rate of BPH eggs, oe6PGDH and WT plants were individually confined in a cylindrical glass cage (diameter 4 cm, height 8 cm, with 48 small holes, diameter 0.8 mm), into which 10 gravid BPH females were introduced. Twenty-four hours later, BPHs were removed. The number of newly hatched nymphs on each plant was recorded daily until no nymph appeared for three consecutive days. The number of unhatched eggs on each plant was then counted under a stereoscopic microscope, and the hatching rate of BPH eggs on oe6PGDH and WT plants was calculated. The experiment was replicated 10 times.

To determine the survival rate of BPH nymphs and fecundity of BPH females on transgenic lines and WT plants, each plant was confined in a cylindrical glass cage (diameter 4 cm, height 8 cm, with 48 small holes, diameter 0.8 mm). Fifteen newly hatched nymphs were released into each cage and allowed to feed on plants. The number of surviving BPH nymphs and emerging adults every day on each plant was recorded until they all became adults. After emergence, pairs of female and male adults were transferred to new plants (one pair per plant), and the number of BPH eggs on each plant was counted under a stereoscopic microscope after 10 days. The two experiments were replicated 10 (for BPH nymph survival experiment) and 30 (for BPH female fecundity experiment) times.

### 4.14. Statistical Analysis

Two-treatment data were analyzed using Student’s *t*-tests. Data from three or more treatment groups were compared using one-way ANOVA followed by Duncan’s multiple range test. Datasets that were not normally distributed or had unequal variance were log- or square-root-transformed before analysis. All statistical analyses were performed using the IBM SPSS Statistics 26 software (IBM Corp., Armonk, NY, USA).

## Figures and Tables

**Figure 1 plants-09-01529-f001:**
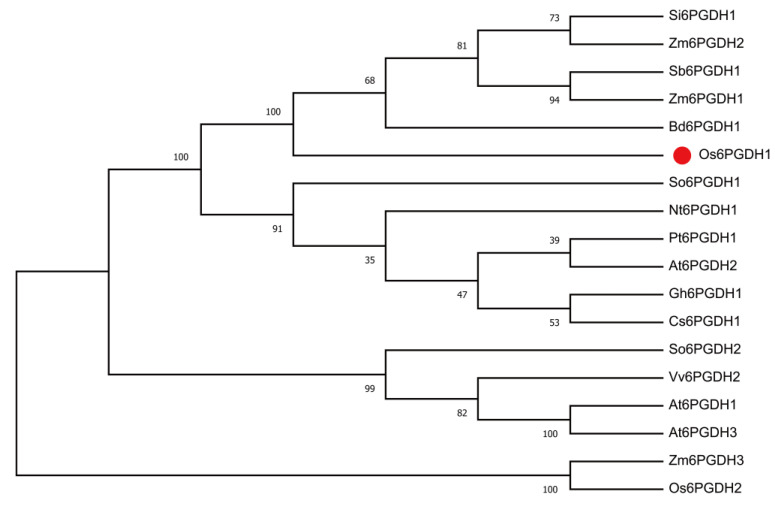
Phylogenetic analysis of 6PGDH proteins from different plant species. The plant species and NCBI accession numbers used in phylogenetic analysis are as follows: *Oryza sativa* Os6PGDH1 (XP_015642949.1), Os6PGDH2 (XP_015616899.1); *Setaria italica* Si6PGDH1 (XP_004971196.1); *Sorghum bicolor* Sb6PGDH1 (XP_021305860.1); *Brachypodium distachyon* Bd6PGDH1 (XP_003557150.1); *Zea mays* Zm6PGDH1 (NP_001266302.1), Zm6PGDH2 (NP_001104910.2), Zm6PGDH3 (NP_001241786.2); *Populus trichocarpa* Pt6PGDH1 (XP_006373519.1); *Arabidopsis thaliana* At6PGDH1 (NP_176601.1), At6PGDH2 (NP_186885.1), At6PGDH3 (NP_198982.1); *Vitis vinifera* Vv6PGDH2 (XP_002275970.1); *Gossypium hirsutum* Gh6PGDH1 (XP_016740720.1); *Cucumis sativus* Cs6PGDH1 (XP_004141482.1); *Nicotiana tabacum* Nt6PGDH1 (AKB09093.1); *Spinacia oleracea* So6PGDH1 (AAK51690.1), So6PGDH2 (AAK49897.1). The Os6PGDH1 is indicated with a red dot ‘●’.

**Figure 2 plants-09-01529-f002:**
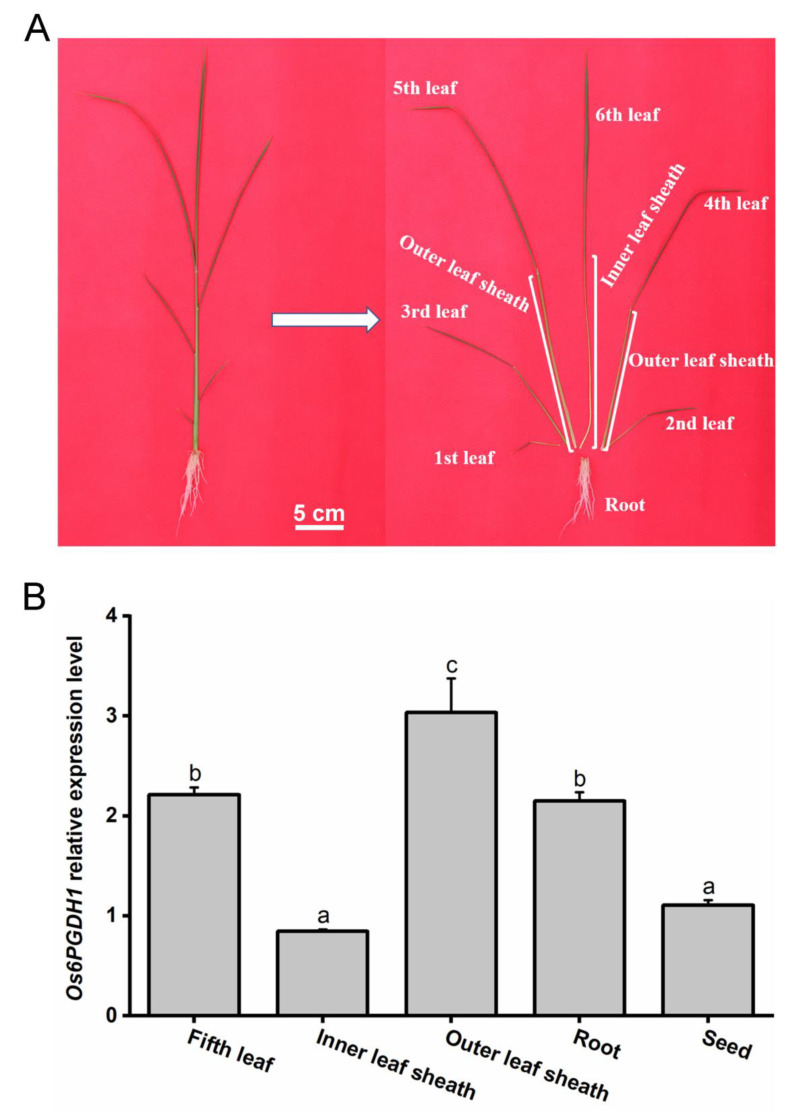
Expression profile of *Os6PGDH1* gene in various tissues of rice. (**A**) Schematic of a 30-day-old seedling with a fully expanded fifth leaf. Samples were collected from the fifth fully expanded leaves, inner leaf sheaths, outer leaf sheaths, roots (tissues indicated), and seeds of rice variety Xiushui 110, and used for RNA extraction and quantitative real-time PCR (qRT-PCR) analysis. (**B**) Relative transcript levels of *Os6PGDH1* in different tissues of rice. Values are means + SE, n = 6. Different letters indicate significant differences among tissues (one-way ANOVA followed by Duncan’s multiple range test, *p* < 0.05).

**Figure 3 plants-09-01529-f003:**
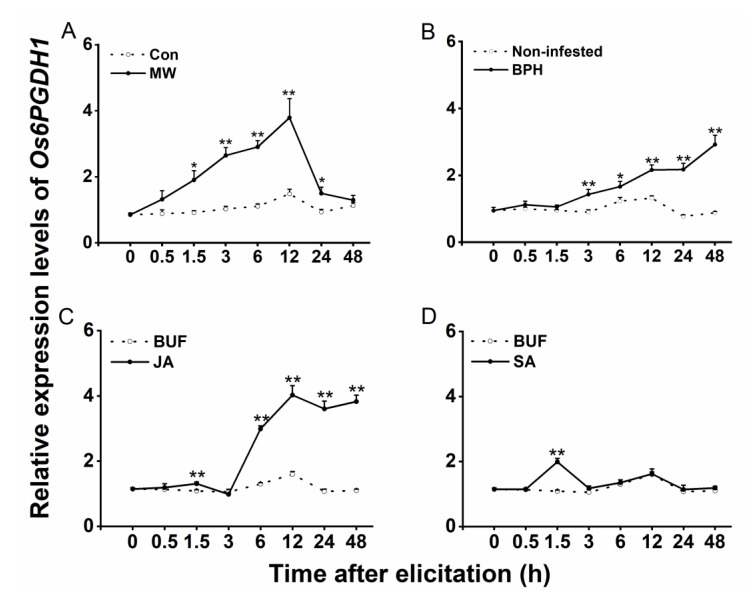
Transcript levels of *Os6PGDH1* in rice after different treatments. Relative transcript levels (mean + SE, n = 5–6) of *Os6PGDH1* in rice outer leaf sheaths that were mechanically wounded (MW) (**A**), infested with gravid BPH females (BPH) (**B**), or treated with JA (**C**) or SA (**D**). Con, nonmanipulated rice plants; BUF, buffer. Asterisks indicate significant differences between treatments and controls (* *p* < 0.05, ** *p* < 0.01, Student’s *t*-test).

**Figure 4 plants-09-01529-f004:**
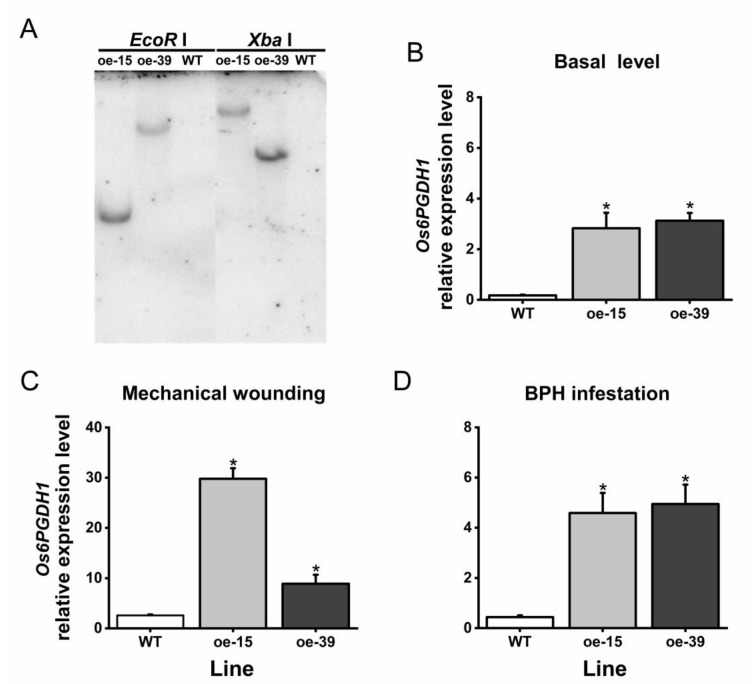
Molecular confirmation of transgenic lines overexpressing *Os6PGDH1*. (**A**) Southern blot of *Os6PGDH1*-overexpressing lines and WT plants. Genomic DNA from oe-15, oe-39, and WT plants was digested with *EcoR* I and *Xba* I, respectively. Digested genomic DNA was hybridized to a specific GUS probe. (**B**) Basal expression levels (means + SE, n = 5) of *Os6PGDH1* in oe6PGDH lines and WT plants; (**C**,**D**) relative transcript levels (means + SE, n = 5) of *Os6PGDH1* in oe6PGDH lines and WT plants that were mechanically wounded or infested by gravid BPH females for 12 h (these experiments were performed separately). Asterisks indicate significant differences in oe6PGDH lines compared with WT plants (one-way ANOVA followed by Duncan’s multiple range test, *p* < 0.05).

**Figure 5 plants-09-01529-f005:**
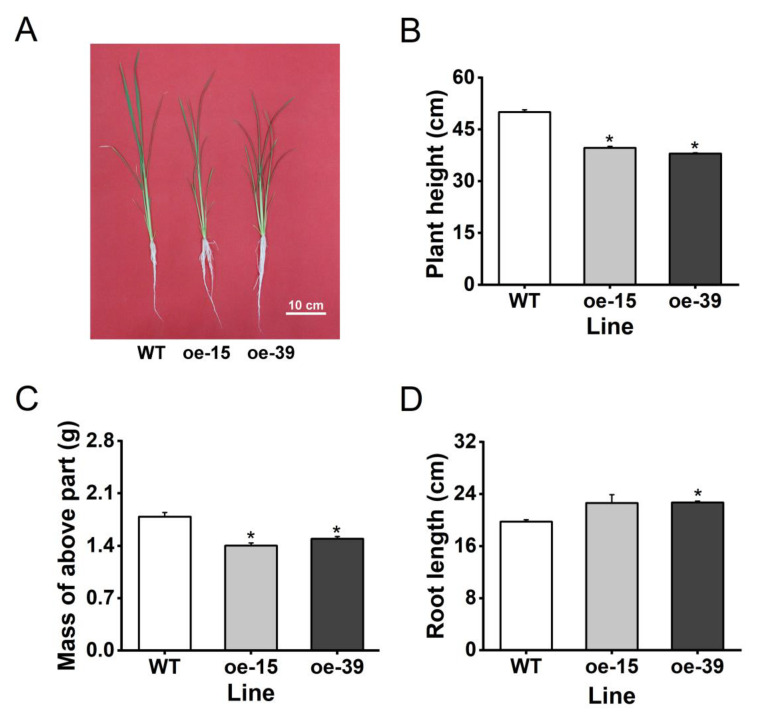
Growth genotypes of oe6PGDH lines and WT plants in greenhouse. (**A**) Growth phenotypes of 30-day-old seedlings of oe6PGDH and WT plants; (B–D) plant height (**B**), mass of aboveground parts (**C**), and root length (**D**) (mean + SE, n = 25) of 30-day-old oe6PGDH and WT plants. Asterisks indicate significant differences in oe6PGDH lines compared with WT plants (one-way ANOVA followed by Duncan’s multiple range test, *p* < 0.05).

**Figure 6 plants-09-01529-f006:**
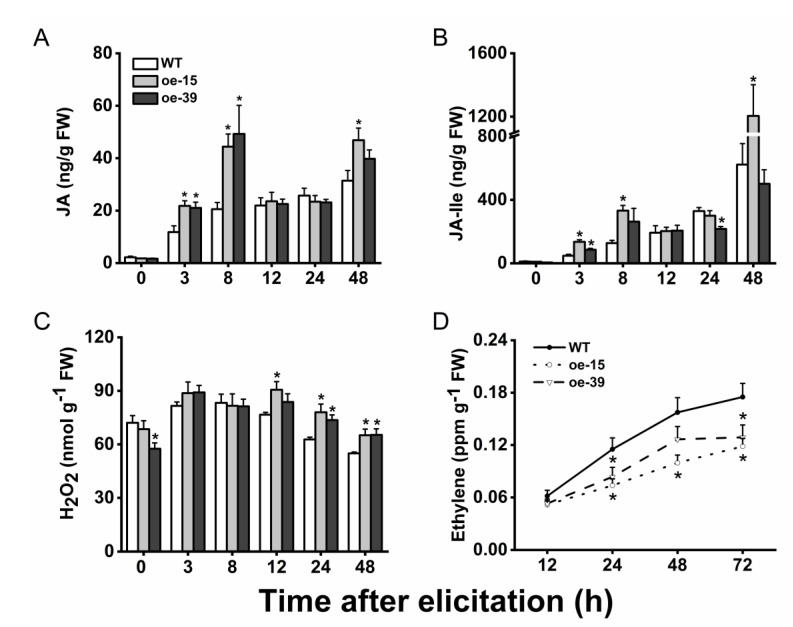
*Os6PGDH1* mediates BPH-induced accumulation of JA, JA-Ile, H_2_O_2_, and ethylene, but not SA. Levels (means + SE, n = 6) of JA (**A**), JA-Ile (**B**), H_2_O_2_ (**C**), and ethylene (**D**) in oe6PGDH lines and WT plants at indicated time points after they were individually infested by 15 gravid BPH females. Asterisks indicate significant differences in oe6PGDH lines compared with WT plants (one-way ANOVA followed by Duncan’s multiple range test, *p* < 0.05).

**Figure 7 plants-09-01529-f007:**
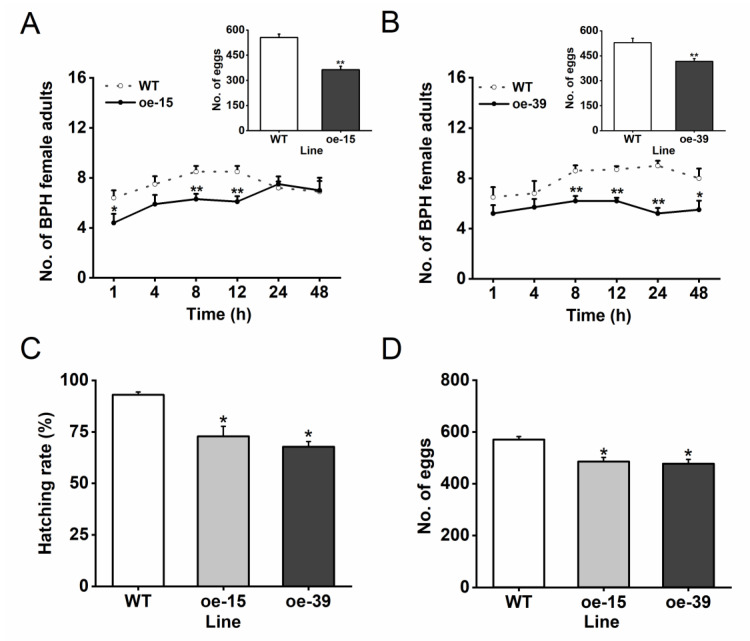
*Os6PGDH1* positively regulates rice resistance to BPH. (**A**,**B**) Mean number (+SE, n = 10) of gravid BPH females per plant on pairs of plants (WT versus oe-15 (**A**) and oe-39 (**B**)). Insert: Mean number (+SE, n = 10) of eggs per plant on pairs of plants as stated above. Asterisks indicate significant differences between oe6PGDH and WT plants (* *p* < 0.05, ** *p* < 0.01, Student’s *t*-test). (**C**) Mean hatching rate (+SE, n = 10) of eggs on oe6PGDH plants and WT plants. (**D**) Mean number (+SE, n = 30) of eggs laid by one pair of BPH female and male adults that emerged from oe6PGDH or WT plants. Asterisks indicate significant differences in oe6PGDH lines compared with those in WT plants (one-way ANOVA followed by Duncan’s multiple range test, *p* < 0.05).

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
