# Peer review of "Overexpression of a Cytosolic 6-Phosphogluconate Dehydrogenase Gene Enhances the Resistance of Rice to Nilaparvata lugens"

_plants, 2020, doi:10.3390/plants9111529_

Round 1

Reviewer 1 Report

The study by Chen et al. describes how the over expression, in rice plants, of a cytosolic 6-phosphogluconate dehydrogenase gene (Os6PGDH1), a central enzyme of the pentose phosphate pathway, (PPP) provided protection from damage caused by the brown plant-hopper (Nilaparvata lugens). As properly mentioned by the authors, several previous reports have described this enzyme´s positive participation in the amelioration of the negative effects causes by diverse abiotic (a)biotic stress condition in several transgenic plant species. The merit of the present study lies in the fact, however, that not much information on the role that the overexpression of this gene might play against insect herbivory by the brown plant-hopper, a highly damaging piercing-sucking insect pest of cultivated rice. Nevertheless, there are several conflicting aspect of this study that must be addressed before it might be ready for publication in Plants. These are described below.

Major concerns:

  1. The authors failed to properly mention how the PPP itself, in addition to several other metabolic pathways that depend on its function, were modified and could have been responsible, not only for altered growth rates observed, but for the protection against N. lugens damage itself. Appropriate mention is made in the “Discussion” section as how altered NADPH/ NADP+ ratios led to stunted growth in Arabidopsis plants in which the activity of NDB1, the external mitochondrial NADPH dehydrogenase, was suppressed, or how could an affected glycolytic pathway may have negatively affected the growth of their Os6PGDH1 overexpressing (OE) plants. This, central, feature of the study under review must be examined in much greater detail. In addition to lower vegetative growth, it is also essential that the authors score the effect that Os6PGDH1 over-expression could have on the seed yield, if this strategy is considered to have a practical application in the future. Also relevant is the possibility that Os6PGDH1 over-expression might have affected plant growth by influencing the active cross-talk known to exist between defense-related phytohormones and other, more growth regulation oriented hormones, such as auxins, cytokinins, gibberellins and brassinosteroids.
  2. In relation to the data presented in Figure 2: What is the physiological relevance of this tissue-specific expression pattern? How does it relate to the objectives/ results of the present study? Why was the expression profile limited to these organs and, apparently, to a single development phase. The authors should consider extending this assay to include the stem panicle and flowers, and other development stages.
  3. In relation to the data presented in Figure 3: Were all treatments compared to the same control? This questions was raised because the expression patterns of the control treatments are all remarkably similar. Please elaborate. The authors are encouraged to investigate the systemic induction of this gene in roots. Please, define if inner AND/ OR outer sheaths were used to obtain these results. The brief, and early, induction of this gene in response to salicylic acid (Fig. 3D) should be explained.
  4. In relation to the data presented in Figure 4: What could be causing that transgenic plants that already are actively expressing Os6PGDH1 in a constitutive way, should further increase its transcript abundance in response to mechanical wounding and/ or brown leaf-hopper herbivory?
  5. The results presented in Figure 5 are explained in a rather facile and experimentally unsupported way by mentioning that they “… indicate that Os6PGDH1 regulates rice growth and development”. In this respect, the authors should consider the concerns raised in point number 1. In addition, they should take into account that, usually, increased carbon and nitrogen requirements for the increased synthesis of defensive compounds or in this case, several other primary metabolism metabolites from the constitutive expression of Os6PGDH1, could have probably imposed a fitness penalty on these transgenic plants.
  6. In relation to the data presented in Figure 7: Please, notice that “antixenosis” might be a better description of the effect observed (refer to Kogan and Ortman, 1978, Antixenosis-a new term proposed to define painter's “nonpreference” modality of resistance. Bulletin of the Entomological Society of America. 24: 175-176). The authors should provide a more detailed explanation of the experimental set-up designed to score the insect´s choice of plants. Did they assure that there were not external factors (e.g., light, color, plant orientation, etc.) that could have affected the preference by the insects of some plants over the others? Plant choice is also known to be influenced by the perception of altered volatile organic compounds (VOCs) composition and/ or the accumulation of toxic secondary metabolite(s) in the plant. Additional experiments designed to answer these possibilities are suggested. Why didn´t the authors perform more straightforward experiments to determine the fitness effect on the insect herbivores, like gain/ loss of weight, retarded development, nymph number, population growth index and/ or increased mortality? Other workers have measured resistance to the brown plant-hopper by measuring this insect´s feeding behavior on different rice cultivars through the use of electrical penetration graphs (EPG), honeydew clocks and/ or number of salivary flanges (refer to Ab Ghaffar et al., 2011, Brown planthopper (N. lugens Stal) feeding behaviour on rice germplasm as an indicator of resistance. PLoS ONE 6: e22137; Quais et al., 2020, Interactions between brown planthopper (Nilaparvata lugens) and salinity stressed rice (Oryza sativa) plant are cultivar-specific. Scientific Reports, 10: 8051). The authors are also encouraged to include the expression of marker genes of the JA and ethylene (ET) defense signaling pathways. Regarding the latter, the inconclusive ET generation data could have been complemented by in vitro assays of ACC synthase activity (refer, for example, to Kende, 1989, Enzymes of Ethylene Biosynthesis. Plant Physiol. 9: 1-4).
  7. In the “Discussion” section, page 11, please define with more precision the presumed positive role that NADPH, generated from the PPP, plays on JA biosynthesis. It is perhaps an indirect effect, via the influence of PPP on the synthesis of fatty acids via pyruvate (refer to Ren et al., 2009, Enhanced docosahexaenoic acid production by reinforcing acetyl-CoA and NADPH supply in Schizochytrium sp. HX-308. Bioprocess and Biosystems Engineering. 32: 837-843). Please elaborate. In this respect, to determine the positive effect observed on the JA-Ile conjugate levels, the authors should perhaps check if a rice homologue of the JAR gene is induced in their Os6PGDH1 over-expressing rice plants.
  8. In the “Discussion” section, page 11, it is stated that “...When attacked by BPH, rice plants perceive the signals from the herbivore and activate the expression of Os6PGDH1, which in turn produces NADPH to promote the biosynthesis of JA, JA-Ile and H2O2”. This statement is unsubstantiated by the experimental data provided in this study. Please delete or soften the argument so that is reads as a possibility rather than a fact.
  9. In section 4.6 of “Material and Methods”, does this way of wounding the plant accurately resemble the way the insects used in this study feed from the plant?

Minor concerns:

  1. In the “Abstract”, what is meant by “… decreased induced levels of ethylene”?
  2. In the “Abstract”, it appears that the JA and JA-ILE pathways might regulate different signaling pathways, without taking into consideration that JA-Ile is one of the biologically active forms of JA.
  3. The following passage in the introduction:“…a double mutant with loss-of-function of the two cytosolic isoforms of G6PDH, G6PD5 and G6PD6 in Arabidopsis, showed increased susceptibility to root knot nematodes due to the suppression of ROS production and gene expression of WRKY transcription factors, SA-responsive and JA-dependent genes” is contradictory. Please consider its elimination or modification.
  4. In Figure 1: There is no need to use fully spelled scientific names repeatedly. Please, use abbreviated name once the fully spelled scientific name has written in the tree. Usually, abbreviated scientific names are used in these figures, while the fully spelled names are described in their respective figure legends. In the legend of Figure 1, please write “red dot”, “full red circle”, etc., besides the symbol used in the tree.
  5. Please justify the use of gravid BPH females in this study.
  6. In the legend of Figure 5, please avoid the repetitive use of the word “in greenhouse”, since this is experimental condition has already been defined in the figure´s title.
  7. In section 4.7, please avoid starting a sentence with a number. Thus, change: “25 individual plants of each line were used for the measurement” to “Twenty five individual plants of each line were used for the measurement”.

8. Check “Reference” list for additional errors to those found in reference No. 6 where the list of authors is incomplete (truncated by the insertion of the short et al. Latin term) and in reference No. 31, in which one of the page number is missing (a + sign appears instead).

Reviewer 2 Report

Review and comments to the manuscript ID plants-967470

Authors: Lin Chen et al.

In my opinion, this manuscript is interesting, and I think that it should be accepted for printing, but not in its current form. Currently, the work requires minor corrections such as:

1) Figs. 1, 2A, 4A (in particular because it is important for research)

2) Minor linguistic and punctuation corrections - please read the works very carefully again.
